# Correlation of Binding and Neutralizing Antibodies against SARS-CoV-2 Omicron Variant in Infection-Naïve and Convalescent BNT162b2 Recipients

**DOI:** 10.3390/vaccines10111904

**Published:** 2022-11-11

**Authors:** Jia Fu, Xiaoying Shen, Mark Anderson, Michael Stec, Tia Petratos, Gavin Cloherty, David C. Montefiori, Alan Landay, James N. Moy

**Affiliations:** 1Division of Allergy and Immunology and Division of Gerontology, Department of Internal Medicine, Rush University Medical Center, Rush University, Chicago, IL 60302, USA; 2Department of Surgery, Duke Human Vaccine Institute, Duke University Medical Center, Durham, NC 27710, USA; 3Abbott Laboratories, Abbott Diagnostics Division, Abbott Park, IL 60064, USA

**Keywords:** SARS-CoV-2, BNT162b2, Omicron, Delta, Beta, variants of concern, binding antibodies, neutralizing antibodies

## Abstract

In vaccine clinical trials, both binding antibody (bAb) levels and neutralization antibody (nAb) titers have been shown to be correlates of SARS-CoV-2 vaccine efficacy. We report a strong correlation bAb and nAb responses against the SARS-CoV-2 Omicron (BA.1) variant in infection-naïve and previously infected (convalescent) individuals after one and two doses of BNT162b2 vaccination. The vaccine-induced bAb levels against Omicron were significantly lower compared to previous variants of concern in both infection-naive and convalescent individuals, with the convalescent individuals showing significantly higher bAb compared to the naïve individuals at all timepoints. The finding that bAb highly correlated with nAb provides evidence for utilizing binding antibody assays as a surrogate for neutralizing antibody assays. Our data also revealed that after full vaccination, a higher percentage of individuals had undetectable Omicron nAb (58.6% in naive individuals, 7.4% in convalescent individuals) compared to the percentage of individuals who had negative Omicron bAb (0% in naive individuals, 0% in convalescent individuals). The discordance between bAb and nAb activities and the high degree of immune escape by Omicron may explain the high frequency of Omicron infections after vaccination.

## 1. Introduction

It has been demonstrated by several studies that in COVID-19 vaccine recipients, both binding antibodies (bAbs) against the SARS-CoV-2 Wild Type (WT) spike protein and neutralizing antibodies (nAbs) against WT virus were correlates of vaccine efficacy [1,2,3,4,5]. However, the level of bAb might not predict virus-neutralizing antibody potency [6]. In addition, the WT virus is no longer in circulation. The SARS-CoV-2 Beta variant (B.1.351) and Delta variant (B.1.617.2) were identified in late 2020, and soon sparked a second and third COVID-19 wave throughout the world and as of December 2021, the SARS-CoV-2 variant Omicron (B.1.1.529) and its subvariants have been the predominant variants of concern (VOC) infecting the United States population [7]. Because Omicron carries at least 15 mutations in the Receptor-Binding Domain (RBD) of the spike protein, which is the major target of nAb [8], Omicron acquires the enhanced affinity for ACE2 and neutralizing immune evasion. This could explain why Omicron has a higher infectivity and higher transmissibility than previous VOCs [9]. In fact, nAb activity against Omicron is 20- to 30-fold lower than against ancestral SARS-CoV-2 [10,11,12]. We previously reported that neutralizing antibodies (nAb) against Omicron are significantly lower than nAb against WT and other VOC [7]. Given the similarities of the bAb and nAb against the WT virus with respect to correlates of protection, we sought to determine if bAb correlated with nAb for Omicron, Beta and Delta VOC. Therefore, we tested stored plasma samples from infection-naïve and previous-infected (convalescent) individuals vaccinated with BNT162b2 in December 2020 and January 2021.

## 2. Materials and Methods

### 2.1. Study Subjects

This study was approved by the Rush University institutional review board. All participants provided written informed consent. All individuals received their first dose of BNT162b2 between December 2020 and January 2021. Available plasma samples from 29 infection-naïve and 27 convalescent individuals who participated in our previously-published studies [2,4] were tested for anti-spike bAb against WT(Wuhan), D614G, Beta (B.1.351), Delta (B.1.617.2), Omicron (B.1.1.529, BA.1) and nAb against D614G, Beta, Delta and Omicron. The samples were collected at 0 to 3 days before vaccination (baseline), 3 weeks after the first dose (T1) and 1 month after the second dose (T2).

None of the 29 individuals in the infection-naïve group had histories of COVID-19 symptoms, positive polymerase chain reaction (PCR) tests, anti-nucleocapsid IgG or anti-Receptor Binding Domain (RBD) IgG at baseline. Among the 27 convalescent individuals, 22 had histories of COVID-19 symptoms and positive PCR tests. The 22 individuals with positive PCR tests were infected between March and November 2020, when D614G was the dominant variant. Among these 22, 18 had positive anti-nucleocapsid and anti-RBD IgG at baseline, and 4 were negative for anti-nucleocapsid and anti-RBD IgG at baseline. Five individuals had positive anti-nucleocapsid IgG detected when they participated in a SARS-CoV-2 antibody screening research study in April and May 2020 but did not have COVID-19 symptoms and did not undergo PCR testing.

All 29 individuals in the infection-naïve group had plasma samples available for testing at all 3 time points. In the convalescent group all 27 individuals had T2 plasma samples tested, 24 had available plasma samples for testing at baseline and 25 had available plasma T1 plasma sample.

### 2.2. Anti-Spike IgG Assay

The bAb were measured with a validated assay using commercial plates manufactured by Meso Scale Discovery (Rockville, MD, USA) provided as an 8-plex, detecting the SARS-CoV-2 prefusion stabilized spike protein of variants including WT, (AY.4.2), (B.1.1.529; BA.1; B.1.15), (B.1.1.7), (B.1.351), (B.1.617.2; AY.3; AY.5; AY.6; AY.7; AY.14) Alt Seq 1, (B.1.617.2; AY.4) Alt Seq 2 and (P.1). Results were analyzed using MSD Discovery Workbench 4.0.12 and reported in WHO/NIBSC International Standard Units (BAU/mL), with values ≥ 17.66 BAU/mL considered positive.

### 2.3. Anti-RBD IgG Assay

Anti-RBD (D614G) IgG levels were measured on an Abbott ARCHITECT i2000SR as described and reported in our previously published study [4]. The results from our previous study were converted from AU/mL to BAU/mL, (with values ≥ 7.1 BAU/mL considered positive) and used in the correlation analyses in this study.

### 2.4. Pseudovirus Neutralization Assay

Neutralizing antibodies were measured as a function of reductions in luciferase reporter gene expression after a single round of infection with SARS-CoV-2 D614G (or Beta or Delta or Omicron) spike-pseudotyped virus in 293T/ACE2 cells as titers as previously described [1,3]. The ID50 nAb titers were calculated based on a dose–response curve. The limit of detection (LOD) was ID50 of 20.

### 2.5. Statistical Analysis

Statistical analysis was conducted with Prism Version 9.2.0 (GraphPad Software, Inc., San Diego, CA, USA) using a 2-tailed nonparametric Mann–Whitney test for comparison between two groups and Spearman’s Correlation to assess the correlations among SARS-CoV-2 anti-spike IgG (BAU/mL), anti-RBD IgG (BAU/mL) and nAb titers (ID50).

## 3. Results

Among the 56 participants, 29 (48.2%) were White individuals and 42 (75%) were female individuals. The mean (SD) age was 42.6 (11.4) years.

Consistent with our previous findings for nAb activity [2], SARS-CoV-2 bAb levels were significantly (all *p* < 0.001) higher in the convalescent group than in the infection-naïve group at all three time points for all the variants tested. The geometric means (95% confidence interval) for SARS-CoV-2 WT anti-spike IgG were 0.80 (0.56–1.1) compared to 28 (11–69) BAU/mL (+35-fold) at baseline; 267 (202–353) compared to 2066 (1026–4160) BAU/mL (+7.74-fold) at T1; and 1345 (1110–1630) compared to 3117 (2306–4213) BAU/mL (+2.32-fold) at T2 in the infection-naïve and convalescent groups, respectively (Figure 1A). Beta anti-spike IgG levels were 0.53 (0.41–0.70) compared to 13 (5.1–33) BAU/mL (+24.53-fold) at baseline; 146 (109–197) compared to 1062 (526–2144) BAU/mL (+7.27-fold) at T1; and 476 (398–570) compared to 1481 (1051–2085) BAU/mL (+7.27-fold) at T2 in the infection-naïve and convalescent groups, respectively (Figure 1B). Delta anti-spike IgG levels were 0.55 (0.40–0.76) compared to 18 (7.1–45) BAU/mL (+32.73-fold) at baseline; 189 (143–251) and 1446 (719–2907) BAU/mL (+7.65-fold) at T1; 688 (581–816) compared to 2085 (1504–2891) BAU/mL (+3.03-fold) at T2 in the infection-naïve and convalescent groups, respectively (Figure 1C). Omicron anti-spike IgG levels were 0.29 (0.21–0.38) compared to 4.6 (2.1–10) BAU/mL (+15.86-fold) at baseline; 41 (30–57) and 379 (179–802) BAU/mL (+9.24-fold) at T1; and 219 (177–272) compared to 585 (417–820) BAU/mL (+2.67-fold) at T2 in the infection-naïve and convalescent group, respectively (Figure 1D).

Figure 1A–D also shows that bAb levels for all the variants increased after each dose of vaccine in the infection-naïve group (all *p* < 0.001), while in the convalescent group bAb levels increased only from baseline to T1 for all variants. In the infection-naïve group, WT bAb increased 333.8-fold from baseline to T1 and 5.04-fold from T1 to T2. The increases from baseline to T1 and from T1 to T2 for Beta were 275.5-fold and 3.26-fold, respectively, for Delta they were 343.6-fold and 3.64-fold and for Omicron they were 141.4-fold and 5.34-fold. In the convalescent group, WT bAb increased 73.8-fold (*p* < 0.001) from baseline to T1 and 1.51-fold (*p* = 0.76) from T1 to T2. The increases from baseline to T1 and from T1 to T2 for Beta were 81.7-fold (*p* < 0.001) and 1.39-fold (*p* = 0.93), respectively, for Delta they were 80.3-fold (*p* < 0.001) and 1.44-fold (*p* = 0.94) and for Omicron they were 82.4-fold (*p* < 0.001) and 1.54-fold (*p* = 0.65).

Figure 1E,F shows that in both the infection-naïve group and the convalescent group, anti-spike IgG levels against Beta, Delta and Omicron were significantly (all *p* < 0.05) lower than anti-spike IgG levels against WT at T1 and T2, with Omicron IgG being the lowest and Beta IgG being the second lowest. For the infection-naïve individuals, Omicron anti-spike IgG levels were 6.51-fold lower (*p* < 0.001) at T1, and 6.14-fold lower (*p* < 0.001) at T2 compared to WT anti-spike IgG levels (Figure 1E). In convalescent individuals, Omicron anti-spike IgG levels were 6.09-fold lower (*p* = 0.002) at baseline, 5.45-fold lower (*p* = 0.002) at T1 and 5.33-fold lower (*p* < 0.001) at T2 compared to WT anti-spike IgG levels (Figure 1F).

Because nAb or bAbs have been demonstrated to be good correlates of mRNA-1273 vaccine protection against COVID-19 [1], we next sought to determine if anti-spike IgG induced by BNT162b2 correlated with anti-RBD IgG and nAbs. Analysis of all data for all subjects at all time points (*n* = 163) showed that anti-spike IgG levels, anti-RBD IgG levels and nAb titers were strongly correlated for WT, Beta, Delta and Omicron, with r values ranging from 0.696 to 0.995 (all *p* < 0.001) (Figure 2A). When we separated the naïve (n = 87) and convalescent groups (*n* = 76), we found that compared to the naïve group, the convalescent group showed stronger correlations among the assays. In the naïve group (Figure 2B), the Omicron nAb assay did not correlate with any of the other bAb or nAb assays (all *p* > 0.05), while in the convalescent group (Figure 2C), all of the assays strongly correlated with each other for WT, Beta, Delta and Omicron with r values ranging from 0.68 to 0.99 (all *p* < 0.001).

Although the bAb levels and nAb titers correlated well, our data indicated that there was a higher percentage of individuals who did not have detectable nAb compared to individuals who had negative bAb levels. Figure 3 compares the percentages of participants who had negative anti-spike IgG and anti-RBD IgG levels, and undetectable nAb titers. At baseline, all 29 naïve (100%) individuals and 4 of 24 (16.7%) convalescent individuals had negative WT anti-spike IgG; all 29 (100%) naïve individuals and six (25.0%) convalescent individuals had negative Beta anti-spike IgG; all 29 (100%) naïve individuals and 5 (20.8%) convalescent individuals had negative Delta anti-spike IgG; and 29 (100%) naïve individuals and 13 (54.3%) convalescent individuals had negative Omicron anti-spike IgG. There was concordance across VOCs for the individuals who had negative IgG levels. After T1, all participants achieved positive anti-spike IgG for WT, Beta and Delta. However, there were 6 (20.7%) naïve individuals and 2 of 25 (8.0%) convalescent individuals who still had negative anti-spike IgG levels for Omicron who later all became positive at T2. All 29 (100%) naïve individuals and 4 (17%) convalescent individuals had negative WT anti-RBD IgG levels at baseline, and all seroconverted after single dose of vaccine.

Compared to the bAb assays, a larger percentage of individuals had an undetectable nAb titer against all the variants at all three time points. At baseline, all 29 (100%) naïve individuals and seven (29.2%) convalescent individuals had undetectable D614G nAb titers; 28 (96.6%) naïve individuals and 20 (83.3%) convalescent individuals had undetectable Beta nAb titers; 28 (96.6%) naïve individuals and 14 (58.3%) convalescent individuals had undetectable Delta nAb titers; and 27 (93.1%) naïve individuals and 19 (79.2%) convalescent individuals had undetectable Omicron nAb titers. After the first dose of vaccine, seven (24.1%) naïve individuals and one (4.0%) convalescent individual had undetectable D614G nAb titers; 20 (69.0%) naïve individuals and 5 (20.0%) convalescent individuals had undetectable Beta nAb titers; 11 (37.9%) naïve individuals and 2 (8.0%) convalescent individuals had undetectable Delta nAb titers; 24 (82.8%) naïve individuals and 3 (12.0%) convalescent individuals had undetectable Omicron nAb titers. Even after two doses of vaccine, there were still 2 (6.9%) naïve individuals who had undetectable Beta nAb titers, and 17 (58.6%) naïve individuals and 2 (7.4%) convalescent individuals who had undetectable Omicron nAb titers (Figure 3).

## 4. Discussion

In this study, we found that two doses of BNT162b2 significantly increased bAb levels to the SARS-CoV-2 Omicron variant in both infection-naive and convalescent individuals, with the convalescent individuals having significantly higher bAb than the infection-naive individuals. Our previous studies [13] and others [14] reported similar effects of BNT162b2 on the bAb levels of the vaccinees. Here we also reported that the BNT162b2 vaccine induced less bAb activity against Omicron compared to WT and the Beta and Delta variants in both infection-naive and convalescent individuals. The BNT162b2 mRNA vaccine was designed based on the original SARS-CoV-2 WT spike protein, and Omicron carries more spike mutations than Beta and Delta. Therefore, the antibody responses against Omicron induced by vaccination were not as robust as against WT, Beta and Delta.

In addition, antibody responses to vaccination alone, SARS-CoV-2 natural infection alone or both (also known as hybrid immunity) have been a topic of discussion. Vaccination induces antibodies with greater binding and neutralization activity to the SARS-CoV-2 Alpha, Beta, Gamma and Delta variants in previously infected compared to infection-naïve individuals [15]. Hybrid immunity is also superior to vaccination alone with regard to symptomatic BA.1 and BA.2 infections [16]. Our study also supports previous findings that the convalescent individuals with hybrid immunity have better binding antibody responses to VOCs compared to individuals with vaccination only. Our results showed that the bAb and nAb assays that were tested in this study strongly correlated with each other, suggesting that bAb test can be used as readily available surrogate tests for nAb assays. Although the clinical relevance of a quantitative bAb result remains unclear due to the lack of a clear definition of a threshold of antibody response sufficient to provide protection against SARS-CoV-2 infection or development of severe disease if infection occurs, there is a positive correlation between antibody levels and protection against COVID-19 according to many studies [1,2,3,4,5]. Since the gold standard assays such as plaque reduction neutralization tests are expensive and time consuming, anti-spike IgG tests could be a good surrogate assay for nAb assays. The advantages of the anti-spike IgG assay are: (1) it is both sensitive and specific for detection of antibody responses to both SARS-CoV-2 infection and vaccination; (2) it allows for an assessment of antibodies against multiple antigenic targets in one well, requiring fewer plasma samples and reagents and less time, and allows for high throughput testing; (3) results are normalized to WHO standardized IU/mL, which enables better quality control, not only among laboratories but also among assays.

However, when we analyzed the correlations among different assays in the naive and convalescent group separately, we found that the strong correlations remained in the convalescent group but not in the naive group. One possible explanation is that in the naive group, a large number of individuals had undetectable nAb titers while having low bAb levels. Therefore, there was no correlation between bAb and nAb at the low end of antibody values. Another possibility is that hybrid immunity might provide higher antibody cross-reactivity against different VOCs compared to the immunity acquired by vaccination only.

Although the bAb and nAb assays correlated well in our study, there was a considerable percentage of individuals who were below the detection levels of the nAb assay compared to the bAb assay. This discrepancy might reflect the sensitivity and dynamic range of the assays. Therefore, we must be cautious because relying only on the bAb assay would miss the individuals who did not have detectable nAb even though they had positive bAb.

Three convalescent individuals in this study did not acquire positive nAb levels against Omicron after the first dose and two of them still remained negative even after the second dose. It appears that a third of dose is necessary for some individuals to achieve positive levels of Omicron nAb. These observations indicate that the reduced antibody responses against Omicron might lead to reduced protection against infection, and perhaps increased transmission.

Limitations of the study include the small sample size and low diversity of participants, in terms of age, sex and race, and therefore, this study does not represent the general population. Since the participants were enrolled in March to November 2020, we presume all the convalescent individuals were infected with D614G variants and our findings could not be applied to other VOC infections. In addition, this study was conducted with samples taken after two doses of BNT162b2 and could not be generalized to other vaccines. Our longitudinal study on these participants after their third and fourth doses of vaccine is still ongoing to evaluate the immunological responses overtime.

## 5. Conclusions

In vaccine clinical trials, both binding antibody (bAb) levels and the neutralization antibody (nAb) titer have been shown to be correlates of SARS-CoV-2 vaccine efficacy. We report a strong correlation of bAb and nAb responses against the SARS-CoV-2 Omicron (BA.1) variant in infection-naïve and previously infected (convalescent) individuals after one and two doses of BNT162b2 vaccination. The induced bAb levels against Omicron were significantly lower compared to previous variants of concern in both infection-naive and convalescent individuals, with the convalescent individuals showing significantly higher bAb compared to the naïve individuals at all timepoints. The finding that bAb highly correlates with nAb provides evidence for utilizing binding antibody assays as surrogates for neutralizing antibody assays. Our data also revealed a higher percentage of individuals with undetectable nAb compared to the percentage of individuals who had negative binding antibody levels. The discordance between bAb and nAb activities and the high degree of immune escape by Omicron may explain the high frequency of Omicron infections after vaccination.

## Figures and Tables

**Figure 1 vaccines-10-01904-f001:**
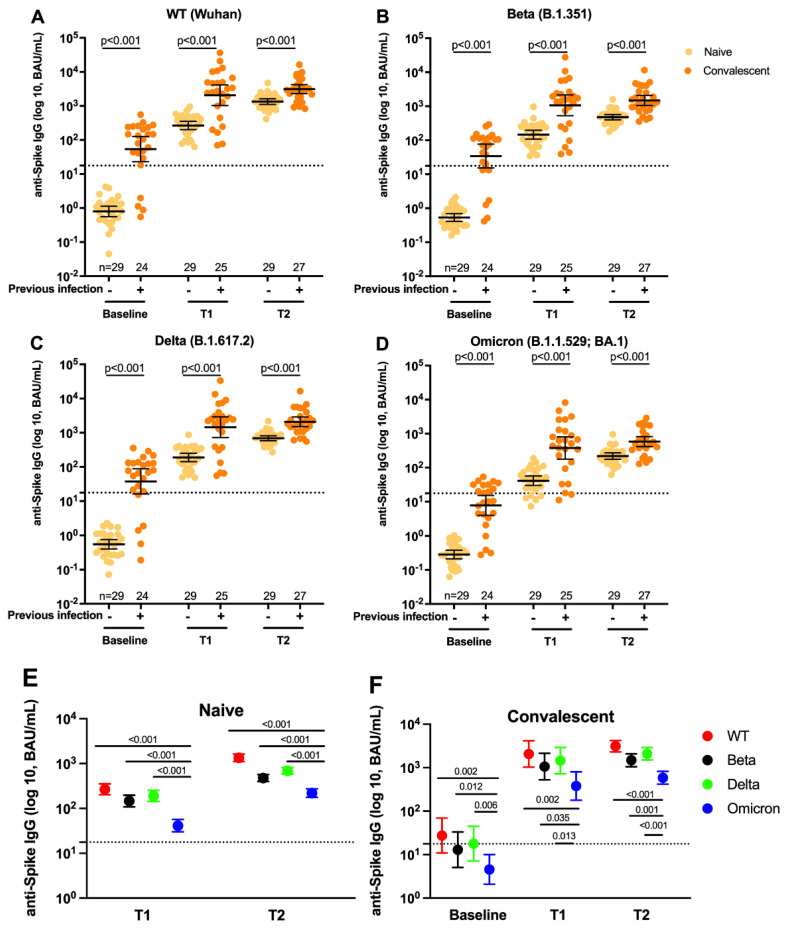
Plasma anti-spike antibody levels against SARS-COV-2 variants, in infection-naïve and convalescent individuals vaccinated with BNT162b2 over time. (**A–D**)**.** Scatter dot plots showing the geometric mean with 95% confidence intervals of the anti-spike IgG levels against Wildtype (**A**), against Beta (**B**), against Delta (**C**), and against Omicron (**D**) at baseline, T1 and T2 in infection-naïve (yellow dots) and convalescent individuals (orange dots). The dash lines represent the assay cutoff at 17.66 BAU/mL. The solid bars correspond to the geometric mean with 95% CI. (**E,F**). Dot graphs showing the geometric means with 95% confidence intervals of anti-spike IgG levels against WT, Beta, Delta and Omicron at T1 and T2 in infection-naïve (**E)** and at Baseline, T1 and T2 in convalescent individuals (**F**). Red dots represent WT anti-spike IgG levels; black dots represent Beta anti-spike IgG levels; green dots represent Delta; blue dots represent Omicron anti-spike IgG levels. Statistical analysis was performed with the 2-tailed nonparametric Mann–Whitney test for comparison between two groups with a *p* < 0.05 considered significant.

**Figure 2 vaccines-10-01904-f002:**
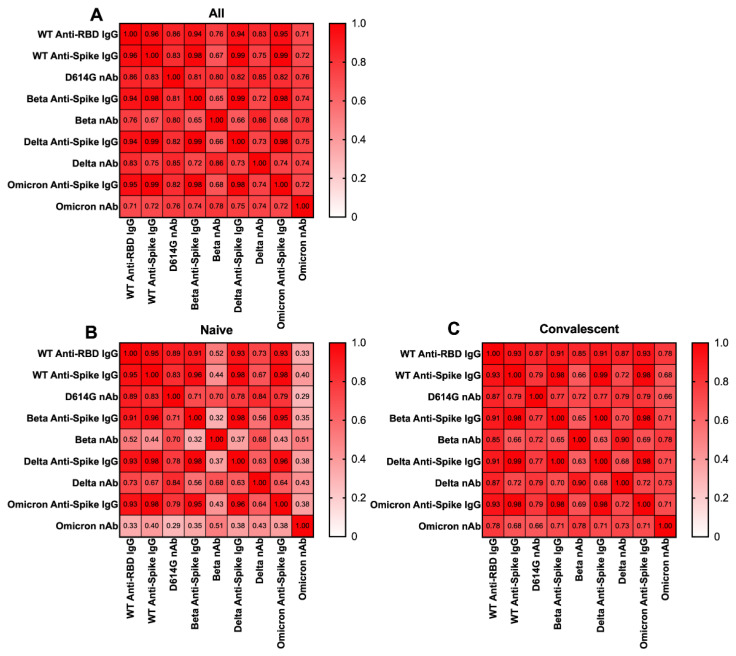
Heatmaps showing the correlations among anti-spike IgG assays, anti-RBD IgG assays and Pseudovirus Neutralization (nAb) assays against WT, Beta, Delta and Omicron. Anti-spike IgG, anti-RBD IgG and nAb data for all individuals (*n* = 163) (**A**), naive individuals (*n* = 87) (**B**) and convalescent individuals (*n* = 76) (**C**) from 3 timepoints were used to perform Spearman’s Correlation to assess correlation between anti-apike IgG levels and nAb titers.

**Figure 3 vaccines-10-01904-f003:**
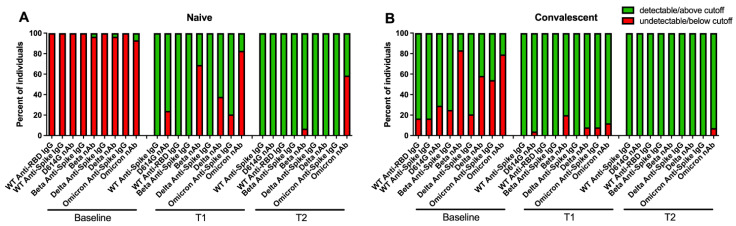
Percentage of individuals below the cutoff values or limit of detection for each SARS-CoV-2 antibody test in naive individuals (**A**) and in convalescent individuals (**B**). The cutoff was at 17.66 BAU/mL and 7.1 BAU/mL for the anti-spike IgG assay and the Anti-RBD IgG assay, respectively. The limit of detection (LoD) of Pseudovirus Neutralization assay was at ID50 of 20. Red bars represent the percentage of individuals below the cutoff or limit of detection. Green bars represent the percentage of individuals above the cutoff or limit of detection.

## Data Availability

The data presented in this study are available on request from the corresponding author.

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
