# Peer review of "Correlation of Binding and Neutralizing Antibodies against SARS-CoV-2 Omicron Variant in Infection-Naïve and Convalescent BNT162b2 Recipients"

_vaccines, 2022, doi:10.3390/vaccines10111904_

Round 1
Reviewer 1 Report
1. In abstract, Our data also revealed a higher..... so in abstract please significant data points
2. Use conventional English/grammar in the manuscript. For eg: It has been demonstrated by several studies that in COVID-19 vaccine recipients, both binding antibodies (bAbs) against the SARS-CoV-2 Wild Type (WT) spike protein and neutralizing antibodies (nAbs) against WT virus were correlates of vaccine efficacy. 2. Given the similarities of the bAb and nAb against the WT virus with respect to correlates of protection, we sought to determine if bAb correlated with nAb for Omicron, Beta and Delta VOC.
3. Also please write something about binding antibodies (bAbs) and neutralizing antibodies in the begining of introduction
4. Shorten the title
5. Results interpretation is not robust
6. Table1 findings can be represented by plots
Author Response
Response to Reviewer 1 Comments
Thank you for reviewing our manuscript and for your helpful comments. Below are our responses to your comments.
- In abstract, Our data also revealed a higher..... so in abstract please significant data points
Response: We have revised lines 22 through 26 to "Our data also revealed that after full vaccination, a higher percentage of individuals had undetectable Omicron nAb (58.6% in naive individuals, 7.4% in convalescent individuals) compared to the percentage of individuals who had negative Omicron bAb (0% in naive individuals, 0% in convalescent individuals).
- Use conventional English/grammar in the manuscript. For eg: It has been demonstrated by several studies that in COVID-19 vaccine recipients, both binding antibodies (bAbs) against the SARS-CoV-2 Wild Type (WT) spike protein and neutralizing antibodies (nAbs) against WT virus were correlates of vaccine efficacy. 2. Given the similarities of the bAb and nAb against the WT virus with respect to correlates of protection, we sought to determine if bAb correlated with nAb for Omicron, Beta and Delta VOC.
Response: We believe that the grammar is correct. We are using “correlates” as a noun and not a verb.
- Also please write something about binding antibodies (bAbs) and neutralizing antibodies in the begining of introduction
Response: We have revised Line 35 to “However, the level of bAb might not predict virus-neutralizing antibody potency[6]. In addition, ” We also discussed the discrepency between bAb and nAb in Line 262 to 267. “Although the bAb and nAb assays correlated well in our study, there was a considerable percentage of individuals who were below the detection levels of the nAb assay compared to the bAb assay. This discrepancy might reflect the sensitivity and dynamic range of the assays. Therefore, we must be cautious because replying only on bAb assay would miss the individuals who did not have detectable nAb even though they had positive bAb.”
- Shorten the title
Response: We changed the title to “Correlation of Binding and Neutralizing Antibodies Against SARS-CoV-2 Omicron Variant in Infection-Naïve and Convalescent BNT162b2 Recipients". There are now 16 words in the title which is in line with the length of other COVID-19 publications in this journal.
- Results interpretation is not robust
Response: We added in results Line 139 to 141 “Anti-Spike IgG levels against Beta, Delta, and Omicron were significantly (all p<0.05) lower than anti-Spike IgG levels against WT at T1 and T2 in both groups, with Omicron IgG being the lowest and Beta IgG being the second lowest."
We also added in discussion Line 226 to 238s “The BNT162b2 mRNA vaccine was designed based on the original SARS-CoV-2 WT Spike protein, and Omicron carries more Spike mutations than Beta and Delta. Therefore, the antibody responses against Omicron induced by vaccination were not as robust as against WT, Beta and Delta.
In addition, antibody responses to vaccination alone, SARS-CoV-2 natural infection alone, or both (also known as hybrid immunity) has been a topic of discussion. Vaccination induces antibodies with greater binding and neutralization activity to the SARS-CoV-2 Alpha, Beta, Gamma and Delta variants in previously-infected compared to infection-naïve individuals.”
- Table1 findings can be represented by plots
Response: We changed Table 1 to Figure 3 with two panels in Line 197.
(See the uploaded Word document for Figure 3)
Figure 3. Percentage of individuals below the cutoff values or limit of detection for each SARS-CoV-2 antibody test in naive individuals (A) and in convalescent individuals (B). The cutoff is at 17.66 BAU/mL and 7.1 BAU/mL for the Anti-Spike IgG assay and the Anti-RBD IgG assay, respectively. The limit of detection (LoD) of Pseudovirus Neutralization assay is at ID50 of 20. Red bars represent the percentage of individuals below the cutoff or limit of detection. Green bars represent the percentage of individuals above the cutoff or limit of detection.

Reviewer 2 Report
The paper by Fu et al makes an interesting correlation binding and neutralising antibodies for the different SARS-CoV-2 VOCs. They find that - reassuringly - the correlation remains quite good. However, as the figures are complicated, it would be worthwhile to make them somehow easier to read (by e.g. having the colour code in the figure legend in figure 1). The heat map in figure 2 is great by may benefit from a better description. In addition, it would be great to make 2 more heat maps, looking at vaccinated only versus convalescent only (if there are sufficient samples). It would certainly be interesting to see whether the cross-reactivity induced by vaccination only, infection only or both is different.
The literature list is awfully short and maybe should contain some mechanistic work as well (e.g. doi: 10.1111/all.15065. Epub 2021 Sep 14).
It has previously been shown (DOI: 10.1056/NEJMoa2208343) that nAb titers after RNA vaccination with both WT and Omicron are only significantly increased against Omicron (vs WT) in individuals without previous infection. Not in those who had previous infections.
The authors avoid discussing this point. As I mentioned before, it would be key to know what previous exposure to infections of which SARS-CoV-2 variants (if they have at all been infected) the tested individuals have had. I only found a short statement on the bottom left of page 8:
"In participants with previous SARS-CoV-2 infection, geometric mean titers were higher after the mRNA-1273.214 booster than after the mRNA-1273 booster against both ancestral SARS-CoV-2 (D614G) and omicron, with geometric mean titer ratios of 1.27 (95% CI, 1.07 to 1.51) and 1.90 (95% CI, 1.50 to 2.40), respectively (Fig. 3 and Tables S6 and S7)". But it seems the authors state the titers were higher, based on a non-significant difference.
The error bars in Fig. 3 are these SD, or SEM? They are extremely small.
Author Response
Thank you for reviewing our manuscript and for your helpful comments. Below are our responses to your comments.
- The paper by Fu et al makes an interesting correlation binding and neutralising antibodies for the different SARS-CoV-2 VOCs. They find that - reassuringly - the correlation remains quite good. However, as the figures are complicated, it would be worthwhile to make them somehow easier to read (by e.g. having the colour code in the figure legend in figure 1). The heat map in figure 2 is great by may benefit from a better description. In addition, it would be great to make 2 more heat maps, looking at vaccinated only versus convalescent only (if there are sufficient samples). It would certainly be interesting to see whether the cross-reactivity induced by vaccination only, infection only or both is different.
Response: We have added the color-coded figure legend in Figure 1 in Line 149 -158 to make it easier to visualize.
(See uploaded Word document for Figures)
Figure 1. Plasma Anti-Spike antibody levels against SARS-COV-2 variants, in infection-naïve and convalescent individuals vaccinated with BNT162b2 over time. A-D. Scatter dot plots showing the geometric mean with 95% confidence intervals of the anti-Spike IgG levels against Wildtype (A), against Beta (B), against Delta (C), and against Omicron (D) at baseline, T1 and T2 in infection-naïve (yellow dots) and convalescent individuals (orange dots). The dash lines represent the assay cutoff at 17.66 BAU/mL. The solid bars correspond to the geometric mean with 95% CI. E-F. Dot graphs showing the geometric means with 95% confidence intervals of anti-Spike IgG levels against WT, Beta, Delta, and Omicron at T1 and T2 in infection-naïve (E) and at Baseline, T1 and T2 in convalescent individuals (F).
We have added separate heatmaps for naive and convalescent individuals. The convalescent group had better correlations among different assays compared to the naive group. We added in Line 167 to 180. “When we separated the naïve (n=87) and convalescent groups (n=76), we found that compared to the naïve group, the convalescent group showed stronger correlations among the assays. In the naïve group (Figure 2B), the Omicron nAb assay did not correlate with any of the other bAb or nAb assays (all p>0.05), while in the convalescent group (Figure 2C), all of the assays strongly correlated with each other for WT, Beta, Delta, and Omicron with r values ranging from 0.68 to 0.99 (all P < 0.001).”
Figure 2. Heatmaps showing the correlations among Anti-Spike IgG assays, Anti-RBD IgG assays and Pseudovirus Neutralization (nAb) assays against WT, Beta, Delta and Omicron. Anti-Spike IgG, Anti-RBD IgG and nAb data for all individuals (n = 163) (A), naive individuals (n=87) (B) and convalescent individuals (n=76) (C) from 3 timepoints were used to perform Spearman’s Correlation to assess correlation between Anti-Spike IgG levels and nAb titers.”
We also added in discussion Line 254 to 261. “However, when we analyzed the correlations among different assays in naive and convalescent group separately, we found that the strong correlations remained in the convalescent group but not in the naive group. One possible explanation is that in the naive group, there was a large number of individuals who had undetectable nAb titers while having low bAb levels. Therefore, there was no correlation between bAb and nAb at the low end of antibody values. Another possibility is that hybrid immunity might provide higher antibody cross-reactivity against different VOCs compared to the immunity acquired by vaccination only.”
- The literature list is awfully short and maybe should contain some mechanistic work as well (e.g. doi: 10.1111/all.15065. Epub 2021 Sep 14).
Response: We thank the reviewer for referring us to the above publication. We found it to be very useful regarding the mechanism of Delta Variant immune escape. We added 4 references in our revised manuscript. (reference # 6,8,15,16)
We edited Line 40 to 45 “Because Omicron carries at least 15 mutations in the Receptor-Binding Domain (RBD) of the Spike protein, which is the major target of nAb[8], Omicron acquires the enhanced affinity for ACE2 and neutralizing immune evasion. This could explain why Omicron has higher infectivity and higher transmissibility than previous VOCs[9]. In fact, nAb activity against Omicron is 20- to 30-fold lower than against ancestral SARS-CoV-2[10-12].”
- It has previously been shown (DOI: 10.1056/NEJMoa2208343) that nAb titers after RNA vaccination with both WT and Omicron are only significantly increased against Omicron (vs WT) in individuals without previous infection. Not in those who had previous infections.
Response: We thank the reviewer for referring us to the above publication (Chalkias, et al). We agree with the reviewer’s comment of the data in the above publication. This publication enrolled participants who received a 4th dose of vaccine (a bivalent vaccine). The participants in our study received only 2 doses of vaccine. In our study, we also found that only the naïve group had higher Omicron, beta, delta and WT antibodies after the 2nd dose, but not the convalescent group. This data is reflected in Line 126 to 136 in our manuscript. I believe that we cannot compare our study with the Chalkias, et al study. We would respectfully request not to comment on this in our manuscript because the immune response after a second dose of vaccine might be different than after a 4th dose that includes Omicron specific mRNA.
- The authors avoid discussing this point. As I mentioned before, it would be key to know what previous exposure to infections of which SARS-CoV-2 variants (if they have at all been infected) the tested individuals have had. I only found a short statement on the bottom left of page 8:
"In participants with previous SARS-CoV-2 infection, geometric mean titers were higher after the mRNA-1273.214 booster than after the mRNA-1273 booster against both ancestral SARS-CoV-2 (D614G) and omicron, with geometric mean titer ratios of 1.27 (95% CI, 1.07 to 1.51) and 1.90 (95% CI, 1.50 to 2.40), respectively (Fig. 3 and Tables S6 and S7)". But it seems the authors state the titers were higher, based on a non-significant difference.
Response: We thank the reviewer for this comment. It seems that the reviewer is commenting on the Chalkias, et al publication when the reviewer states “I only found a short statement on the bottom left of page 8”. Therefore, we are not sure how to respond to this comment. We did state in our manuscript (Line 54 in Materials and Methods and Line 270 in the Discussion) that our previously infected participants were infected between March to November 2020 and therefore were most likely infected with D614G (although we did not have any sequencing data).
- The error bars in Fig. 3 are these SD, or SEM? They are extremely small.
Response: We are wondering if the reviewer meant figure 1? The error bars are 95% confidence intervals of the geometric means. We have changed Line 147 to "A-D. Scatter dot plots showing the geometric mean with 95% confidence intervals of the anti-Spike IgG levels against Wildtype (A), against Beta (B), against Delta (C), and against Omicron (D) at baseline, T1 and T2 in infection-naïve (yellow dots) and convalescent individuals (orange dots). The dash lines represent the assay cutoff at 17.66 BAU/mL. The solid bars correspond to the geometric mean with 95% CI. E-F. Dot graphs showing the geometric means with 95% confidence intervals of anti-Spike IgG levels against WT, Beta, Delta, and Omicron at T1 and T2 in infection-naïve (E) and at Baseline, T1 and T2 in convalescent individuals (F).”

Reviewer 3 Report
It is an interesting study, but with some limitations, as the authors indicated.
The number of participants is small, the samples are not recent and the response to only one vaccine is studied. However, the data obtained provide some more information about vaccination against SARS-CoV-2.
Author Response
Response to Reviewer 3 comments
Thank you for reviewing our manuscript and for your helpful comments. Below are our responses to your comments.
Comment 1: It is an interesting study, but with some limitations, as the authors indicated.The number of participants is small, the samples are not recent and the response to only one vaccine is studied. However, the data obtained provide some more information about vaccination against SARS-CoV-2.
Response: We agree with what reviewer 3 commented. Our study cannot be applied to other vaccines or other VOC infections and the reviewer pointed out that we stated as such in discussion Lines 276 and 278. We stated in line 275 “Limitations of the study include the small sample size and low diversity of participants, in terms of age, sex, and race;” and we added “and therefore, this study does not represent the general population.” in line 276. Regarding the reviewer’s comment that the samples were not recent. We used these samples because we wanted to evaluate the antibody levels before and after vaccination. The samples were collected at the initial rollout of the COVID-19 vaccines. It would be difficult to collect new samples now because it would be hard to find participates who have not been vaccination and are willing to be vaccinated now.
Reviewer 4 Report
The submitted manuscript deals with antibody responses whereas the special issue is about T cell responses. I suggest that the manuscript will be submitted to a more adequate journal
Author Response
Response to Reviewer 4 comments
Thank you for reviewing our manuscript and for your helpful comments. Below are our responses to your comments.
Comment 1: The submitted manuscript deals with antibody responses whereas the special issue is about T cell responses. I suggest that the manuscript will be submitted to a more adequate journal
Response: Immediately after we submitted our manuscript, we received the following email from the editorial office of Vaccines that stated:
“Submitted to section: COVID-19 Vaccines and Vaccination,
https://urldefense.com/v3/__https://www.mdpi.com/journal/vaccines/sections/COVID-19_vaccines_vaccination__;!!OlavHw!4HtTeGuYnW1FVo4I3eq-dWKID62tPG-Na9saRoHzUR07hIqoRwLeVyy3uYbqj7h1XOkv8-2f58t9_g$
Antibody Response of Vaccines to SARS-CoV-2
https://urldefense.com/v3/__https://www.mdpi.com/journal/vaccines/special_issues/vaccines_antibody__;!!OlavHw!4HtTeGuYnW1FVo4I3eq-dWKID62tPG-Na9saRoHzUR07hIqoRwLeVyy3uYbqj7h1XOkv8-3UXt9sJA$ ”
We did would like our manuscript to be considered for the Special Issue "Antibody Response of Vaccines to SARS-CoV-2" and not the issue on T cell responses.
Round 2
Reviewer 1 Report
authors addressed all the comments